# IMPROVING ACTION CLASSIFICATION WITH BRAIN-INSPIRED DEEP NETWORKS

## ABSTRACT

Recognizing actions from visual input is a fundamental cognitive ability. Perceiving what others are doing is a gateway to inferring their goals, emotions, beliefs and traits. Action recognition is also key for applications ranging from robotics to healthcare monitoring. Action information can be extracted from the body pose and movements, as well as from the background scene. However, the extent to which deep neural networks make use of information about the body and information about the background remains unclear. In particular, since these two sources of information may be correlated within a training dataset, deep networks might learn to rely predominantly on one of them, without taking full advantage of the other. Unlike deep networks, humans have domain-specific brain regions selective for perceiving bodies, and regions selective for perceiving scenes. The present work tests whether humans are thus more effective at extracting information from both body and background, and whether building brain-inspired deep network architectures with separate domain-specific streams for body and scene perception endows them with more human-like performance. We first demonstrate that deep networks trained using the Human Atomic Actions 500 dataset perform almost as accurately on versions of the stimuli that show both body and background and on versions of the stimuli from which the body was removed, but are at chance-level for versions of the stimuli from which the background was removed. Conversely, human participants (N=28) can recognize the same set of actions accurately with all three versions of the stimuli, and perform significantly better on stimuli that show only the body than on stimuli that show only the background. Finally, we implement and test a novel deep network architecture patterned after domain specificity in the brain, that utilizes separate streams to process body and background information. We show that 1) this architecture improves action recognition performance, and 2) its accuracy across different versions of the stimuli follows a pattern that matches more closely the pattern of accuracy observed in human participants.

## 1 INTRODUCTION

Action recognition from videos has a wide variety of applications, in fields as disparate as robotics, surveillance, healthcare, and elderly assistance. It is also a key cognitive ability for humans: much of our lives are spent in the company of others, and we need to understand what they are doing in order to react appropriately. The human brain is able to recognize actions with exceptional accuracy and robustness, and as such it can be a source of inspiration for artificial systems for action recognition. When comparing the human brain to current deep neural networks for action recognition, one key difference stands out. Most deep neural networks process jointly the body that is performing the action and the surrounding background, within the same processing stream. By contrast, the human brain contains regions that respond selectively to bodies (Downing et al., 2001), and separate regions that respond selectively to scenes (Epstein & Kanwisher, 1998). In the present work, we offer three novel contributions. First, we probe the human ability to recognize actions given information about the body, the background, or both. Second, we test the ability of deep networks to classify actions given information about the body, the background, or both. Finally, we implement deep networks inspired by the category-selective architecture observed in the human brain, and test whether this

architecture confers unique advantages for action recognition by comparing their performance to models that process the body and background jointly.

Within a video, the body and the background convey partially redundant information about the category of an action. Many actions tend to occur in typical contexts, for example people cook in a kitchen, and play basketball in a basketball court. Therefore, deep networks might learn to take advantage of the information contained in the background, and neglect to use information about the body. This could worsen performance on independent test data, especially if testing videos contain out-of-distribution backgrounds. In this case, even if the bodies and their movements in the testing data are within the distribution of those in the training data, the networks would fail to recognize the actions in the context of their new backgrounds.

By contrast, processing information about the body and about the background using separate streams could lead deep networks to make the most of both background and body information, thus improving generalization performance. A loss function that includes a term for the performance of the body and background streams combined, but also terms for the performance of the body stream and of the background stream in isolation, would force the network to make the most of information about the body and about the background. Indeed we find that deep networks with this category-selective architecture yield improved accuracy for action recognition. In addition, the pattern of performance of these networks for different types of stimuli (body only, background only, or both) is more similar to the pattern of performance observed in human participants.

## 2 RELATED WORKS

The category-selective approach we propose is related to Dropout (Srivastava, 2013; Baldi & Sadowski, 2013; Labach et al., 2019), in that it computes and uses the loss obtained from different sub-networks. However, it would be very unlikely to obtain sub-networks that use only information about the body or only about the background using Dropout, therefore the category-selective approach is needed to guarantee lack of redundancy between background and body information. Compared to Dropout, the sub-networks in the category-selective approach are fixed, while in Dropout different sub-networks are used in each batch, further reducing redundancy. However, the simultaneous use of separate losses for the different streams as well as for the combined performance enables the category-selective architecture to obtain some of the benefits of redundancy reduction without sacrificing performance on the training data. Since the output of the full network is always computed during training, the category-selective approach more suitable for continuous learning. Recent work has used pose-estimation to obtain strong action recognition performance (Song et al., 2021; Rajasegaran et al., 2023). Here we test whether simpler feedforward convolutional networks, that do not require pose estimation, are able to improve over competing models by leveraging brain-inspired category-selective processing. Previous research in deep learning has taken inspiration from the architecture of human vision, building two-stream networks for video processing (Simonyan & Zisserman, 2014; Feichtenhofer et al., 2019) inspired by the distinction between the ventral and dorsal streams in the human brain (Ungerleider, 1982; Milner & Goodale, 2008). The present work extends this research to include category-selective organization, providing another example of how brain-inspired neural network architectures can yield computational benefits.

Recent models of action recognition have obtained accurate performance used transformer architectures (Li et al., 2022; Ryali et al., 2023). The category-selective architecture introduced in this article could be integrated with transformers, for example by using separate transformer streams to process the body and the background. Accurate performance has also been achieved with models based on self-supervised learning, such as masked autoencoders (Wang et al., 2023a; He et al., 2022) and masked video distillation (Wang et al., 2023b). The current work can be viewed as a form of masked supervised learning, in which the masks are semantically defined objects (the body and the background). Finally, foundation models have been used to achieve state-of-the art action recognition performance (Wang et al., 2022; Xu et al., 2023). While the use of multimodal inputs is beyond the scope of the present article, extending category-selective architectures to multi-modal data is a potential direction for future research.

Due to its implications for object representations in humans, this work is closely related to recent models of the causes of category selectivity in the brain (Arcaro & Livingstone, 2021; Doshi &

Konkle, 2023; Prince et al., 2023), and offers an alternative account of the large-scale organization of ventral temporal cortex.

## 3 METHODS

### 3.1 TRAINING DATASET

In order to facilitate the comparison between human data and the models' performance, the Human-centric Atomic Actions 500 dataset (Chung et al., 2021) was used for the training and testing of the deep neural networks as well as for the behavioral experiments with human participants. The dataset consists of 500 action classes, with 20 videos in each class. We first used YOLO v8 (Redmon et al., 2015) to segment bodies and backgrounds for each frame of all videos. We next selected a subset of the videos that met the following criteria: 1) There was only one person in a video 2) Segmented data passed manual QA inspection (no obvious artefacts). Action categories with fewer than 15 usable videos were discarded. The resulting dataset consisted of 109.089 frames from 121 categories (95056 for training, 6336 for validation, 7697 for testing). As a result of the segmentation procedure, each frame was available in three versions: original ("ORIG", consisting of both background and body), body-only (in which pixels in the background were set to be black), and background-only (in which the body was removed, masked, and inpainted). To mask the silhouette of the body in background-only frames, we summed the body masks across all frames of a video, dilated them by a factor of 1.2, inpainted the resulting region for the entire video (see example stimuli in the appendix, figure S1). This way, the inpainted area does not change frame-to-frame and does not contain an outline (silhouette) that could convey information about body-pose. This was done to mitigate the residual body information in background-only frames. For each of the three versions of the training dataset (original frames, background-only frames and body-only frames) optic flow was computed using previously published MotionNet model Zhu et al. (2018). This model takes in a batch of sequential frames and can be trained to predict a 2D map of optic flows (see figures S2 and S3 in the appendix for examples). The MotionNet model was trained for 5 epochs using batches of 11 frames, taken from the 95056 training frames extracted from the HAA500 dataset.

### 3.2 MODELS OF ACTION RECOGNITION

We evaluated the performance of two different types of neural network architectures. First, we tested neural networks that process entire video frames as input (the "Baseline" networks). Next, we tested networks that have separate processing streams for the body and the background (the "DomainNet" networks). For each network type, we tested two versions: a version that relies exclusively on static features ("frames"), and a version that relies on both static features and optic flow ("frames+flow"). All networks utilized a ResNet-50 architecture He et al. (2015) as the backbone, and were trained to predict an action class (n=121). The DomainNet architecture consisted of two-streams of ResNet-50 (body-only stream and background-only stream), receiving as input body-only frames and background-only frames respectively. Outputs of the final layer of the two streams was then summed (fusion layer) to generate combined predictions and calculate the combined loss. Importantly, the overall loss function of the network was calculated as the sum of three Cross-Entropy loss terms: one based on the output of the background-only stream in isolation, one based on the output of the body-only stream in isolation, and one based on the summed outputs of the two streams:

$$\mathcal{L} = \mathcal{L}_{\text{body}} + \mathcal{L}_{\text{background}} + \mathcal{L}_{\text{combined}} \tag{1}$$

In order to minimize this overall loss function, it is not sufficient for the network to achieve low cross-entropy loss for the combined outputs of the two streams. The cross-entropy loss for each individual stream must be minimized too. As a consequence, even if the output of one stream is sufficient to achieve accurate action recognition on the training data, yielding low values of the loss $\mathcal{L}_{\text{combined}}$, the loss term computed using the other stream in isolation can continue to drive learning within that stream. Only when the individual performance of both streams is optimized, the loss is at its lowest.

### 3.3 TRAINING PROCEDURE

Networks were trained for 100 epochs, with 500 batches per epoch and 32 stimuli per batch. Because some categories contained more frames than others, to mitigate class imbalance, for each image in a batch, we first uniformly sampled a category, and then uniformly sampled a frame and/or a an optic flow belonging to that category. In this way, each action category was sampled an equal number of times during training. The loss function $\mathcal{L}$ was optimized using stochastic gradient descent (SGD), with learning rate of $0.001$ and momentum of $0.9$.

### 3.4 BEHAVIORAL TESTING

#### 3.4.1 PARTICIPANTS

The study was approved by Boston College's Institutional Review Board. A total of N=30 participants completed the online experiment hosted on Amazon Mechanical Turk and were compensated for their time. Two participants did not complete the whole task and were removed from further analyses.

#### 3.4.2 BEHAVIORAL EXPERIMENT

Participants were shown a subset of the Training Dataset videos. Due to time constraints in the behavioral experiment, we selected the 50 action categories that were classified most accurately by the Baseline-frames model using validation data, in order to ensure that any failures of the Baseline model would not be due to the particular action classes used in the study. Human participants were shown one video for each category (for a total of 50 videos). Each video was shown in three versions: original, background-only and body-only. Videos were shown one at a time, and different video types were shown in separate blocks. Within each block, stimuli and answer choices were randomized. To minimize order effects, and minimize the risk that viewing the original videos might affect performance during background-only and body-only trials, participants were first shown the background-only block, followed by body-only, followed by original videos. Each video was presented once for each participant. For each video participants selected one action category among five possible choices (including the correct answer and four randomly selected foils).

### 3.5 ACCURACY METRICS

All models were trained to perform a 121-way classification for each frame in a video. In order to facilitate comparing between model and human accuracies, we report human-aligned accuracies by only considering the 50 categories that were used in the behavioral experiment, averaging network outputs for all frames in a given video and computing the argmax of the networks' outputs among the 5 answer choices that were available to human participants. 121-way classification accuracies (top1 and top5 accuracies) are available in the appendix (tables S1 & S2).

## 4 RESULTS

### 4.1 BEHAVIORAL RESULTS

Human participants were highly accurate (M=98%, SD=2%) at recognizing action when viewing the original videos (consisting of both background and body information). Performance was similarly high when viewing body-only videos (M=94%, SD=.3%). However, the accuracy of human participants dropped substantially when viewing background-only videos (M=76% SD=7%). Participant accuracy was significantly higher when viewing body-only videos than when viewing background-only videos $t(27) = 13.27, p < 0.001$ (Figure 1, Table 1).

### 4.2 MODELLING RESULTS

### 4.3 BASELINE MODELS

Human-aligned accuracy results for Baseline-frames model were: 57% when tested with original frames, 40% when tested with background-only and 20% when tested with body-only version of

the stimuli (chance level 1/5 = 20%). In other words, we observed above chance performance for original and background-only frames and a decrease to chance-level performance for body-only frames. Adding optic flow information (Baseline:frames+flow model) resulted in modest gains in test-accuracy (by 5% for original videos, 7.5% for background-only and 2.5% for body-only, table 1). Overall the pattern of results remained the same, with high accuracies for original and background-only frames+flows and near-chance performance for body-only frames+flows. These results suggest that networks trained using the original frame stimuli (that show both body and background) learn representations of the background but fail to extract information about actions from the body.

### 4.4 DOMAIN-SPECIFIC MODELS

DomainNet:frames model was more accurate than the Baseline:frames model when tested using frames showing both the body and the background ("ORIG", 66.25%). The DomainNet: frames model achieved a 13.75% increase in accuracy over the Baseline:frames model. The DomainNet: frames model also outperformed the Baseline:frames model on the background-only stimuli (achieving an accuracy 42.5%: a 2.5% increase compared to the Baseline:frames model). Importantly, the DomainNet:frames model exhibited high accuracy even when tested using body-only stimuli (62.5%) yielding to a 42.5% increase in accuracy compared to the Baseline:frames model. Like human participants, and unlike the Baseline:frames model, the DomainNet:frames model showed superior performance for body-only inputs than for background-only inputs (figure 1).

Adding optic flow information resulted in substantial gains in accuracy when testing with body-only frames+flows (73.75% – a 11.25% increase over the DomainNet:frames model), with smaller gains when testing with combined frames+flows (75% accuracy, a 8.75% increase) and a somewhat surprising decrease when testing with background-only frames+flows (38.75% accuracy, a 3.75% decrease). Inspecting the 121-way classification results (Appendix), however, reveals that in the 121-way classification the DomainNet:frames+flows model outperforms all other models also for background-only stimuli.

Table 1: Accuracies (in percentage points) of networks compared. Chance level is 20%.

| Network-name | ORIG | body-only | background-only |
|---|---|---|---|
| Baseline: frames | 52.50 | 20.00 | 40.00 |
| Baseline: frames+flows | 57.50 | 22.50 | **47.50** |
| DomainNet: frames | 66.25 | 62.50 | 42.50 |
| DomainNet: frames+flows | **75.00** | **73.75** | 38.75 |
| Humans: | 98.43 | 93.93 | 76.29 |

## 5 DISCUSSION

We tested action recognition performance in humans and artificial neural networks given different inputs: the body, the background, and their combination. Our contribution is threefold: the results 1) quantify the importance of background and body features for human action recognition, 2) show that Baseline deep networks trained using the entire frames (showing both body and background) produce patterns of performance that strongly diverge from those observed in humans, and 3) demonstrates that brain-inspired architectures improve accuracy and yield more human-like patterns of performance.

Humans are very accurate at recognizing actions across a variety of scenarios. In a controlled experiment, we demonstrate that human action recognition remains high even when background or body information is removed. Crucially however, body information seems to be most important, as removing this information resulted in a significant drop in recognition accuracy (from 93% to 76%). Conversely, removing background information resulted in a more modest drop in recognition accuracy. These results suggest that humans do indeed combine multiple sources of information, but that body pose and kinematics are sufficient to achieve high accuracy. The co-occurrence between context and action is not perfect: many actions can be performed in a variety of different settings.

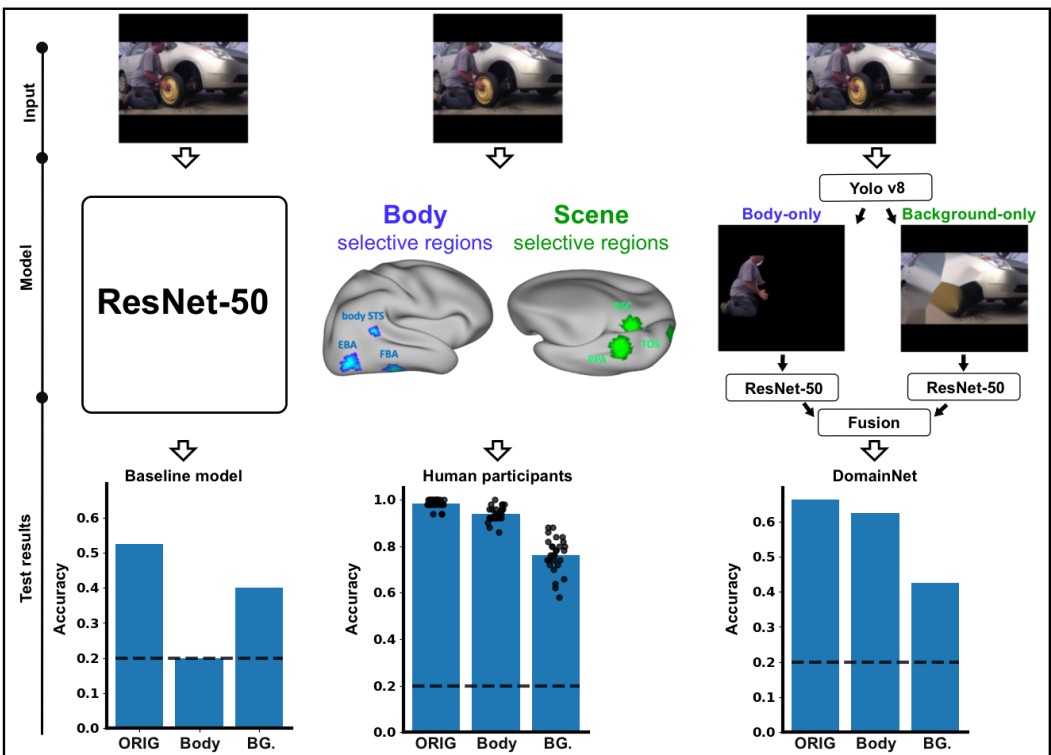

Figure 1: Comparison between network trained using original frames (Baseline:frames model), human results and brain-inspired two stream network (DomainNet:frames). Baseline:frames model performs similarly well when tested using original (ORIG) or background-only (BG.) frames, and at chance-level when tested using body-only (Body) frames. Conversely human participants (N=28) perform similarly well when tested using original and body-only stimuli, with lower performance during background-only trials. Our architecture, patterned after domain-specific pathways in the brain - exhibits 1) higher accuracies across all versions of the stimuli (original, background-only, body-only) and 2) Performs similarly well when tested using original or body-only frames similar to human participants. Note: In order to facilitate results comparison, network accuracy was calculated considering only the categories and answer choices that were available to human participants (50 categories, 5 answer choices).

Therefore, context can help inform action recognition, but the body and its movements provide more accurate information.

The pattern of accuracies observed in human participants was in stark contrast to the performance of Baseline models (Baseline:frames and Baseline:frames+flows). These models, trained on the original frames (showing both body and background), mostly learn representations of the background, and do not generalize well to stimuli that show only the body. A possible reason for this is that pose information represents only a small portion of the input in terms of surface area, and that therefore Baseline deep networks learn background features more rapidly. This could result in sub-optimal performance in situations where an action is being performed in an unlikely context, or when background information is missing.

Inspired by domain-specific pathways in the brain, we tested a novel architecture that processes background and body information in separate streams. This architecture performs better than models trained using the original frames (showing both body and background). These results cannot be explained by Domain-specific models having more parameters: the DomainNet:frames model (which consists of two ResNet-50 streams) reaches higher overall accuracy than the Baseline:frames+flows model (which also consists of two ResNet-50 streams).

A potential criticism of the category-selective architecture introduced in this article could center on the observation that this type of architecture requires the use of a segmentation model (in the present work, we used YOLO Redmon et al. (2016)). However, it is not implausible that the brain might perform object segmentation at stages of processing that precede the emergence of category-selectivity. Such segmentation could rely on "Spelke object inference" (Chen et al., 2022; Spelke, 1990), using static as well as motion features that can be available already at the level of areas V4 and V5 (that is, at earlier processing stages than category-selective regions).

The category-selective organization of object representations in the human brain is a central topic of study in Neuroscience (Mahon & Caramazza, 2009; Peelen & Downing, 2017; Ratan Murty et al., 2021; Dobs et al., 2022). Distinct brain regions show selectivity for faces Kanwisher & Yovel (2006), bodies Downing et al. (2001), scenes Kamps et al. (2016), and objects Beauchamp & Martin (2007); Kaiser et al. (2016). Recent theories of the causes of category selectivity have focused on the role of input statistics in driving the topography of visual representations (Arcaro & Livingstone, 2021; Doshi & Konkle, 2023; Prince et al., 2023). Here we propose an alternative account, according to which category selectivity does not result due to the statistics of the input, but as an adaptation to improve performance at downstream behavior.

This proposal is related to previous evolutionary accounts of the emergence of category-selectivity (Caramazza & Shelton, 1998). In addition, it borrows from recent work investigating how multiple task demands can shape the structure of optimal deep network models (Zamir et al., 2018). Reporting improved action recognition performance in category-selective deep networks that process information about the body and background in separate streams, the present work offers an example of how category-selective organization might be driven by constraints not at the level of the inputs (image statistics), but at the level of the outputs. Constraints at the level of the outputs might lead to multiple aspects of the organization of visual representations, including the separation between ventral, dorsal, and lateral streams (Ungerleider, 1982; Pitcher & Ungerleider, 2021) as well as separate processing streams for different object categories. Importantly, this account can also explain a key fact of the category-selective organization of object representations which cannot be easily reconciled with theories based on the statistics of the inputs: the observation that category-selective organization is also found in congenitally blind participants (Mahon et al., 2009; Striem-Amit & Amedi, 2014; Ratan Murty et al., 2020).

This proposal has implications for the study of visual representations in the human brain, but also for the development of deep neural networks. In fact, to the extent to which the architecture of the brain is sculpted by evolutionary pressures to optimize perception, Neuroscience can offer cues for the development of more powerful systems for artificial vision.

## 6 REPRODUCIBILITY

Code for models used in this paper (Baseline:frames, Baseline-frames+flows, DomainNet:frames, DomainNet:frames+flows), together with data and code used to generate figures can be found at this anonymized link: https://anonymous.4open.science/r/pub-ICLR24-8A46/

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

# A  APPENDIX

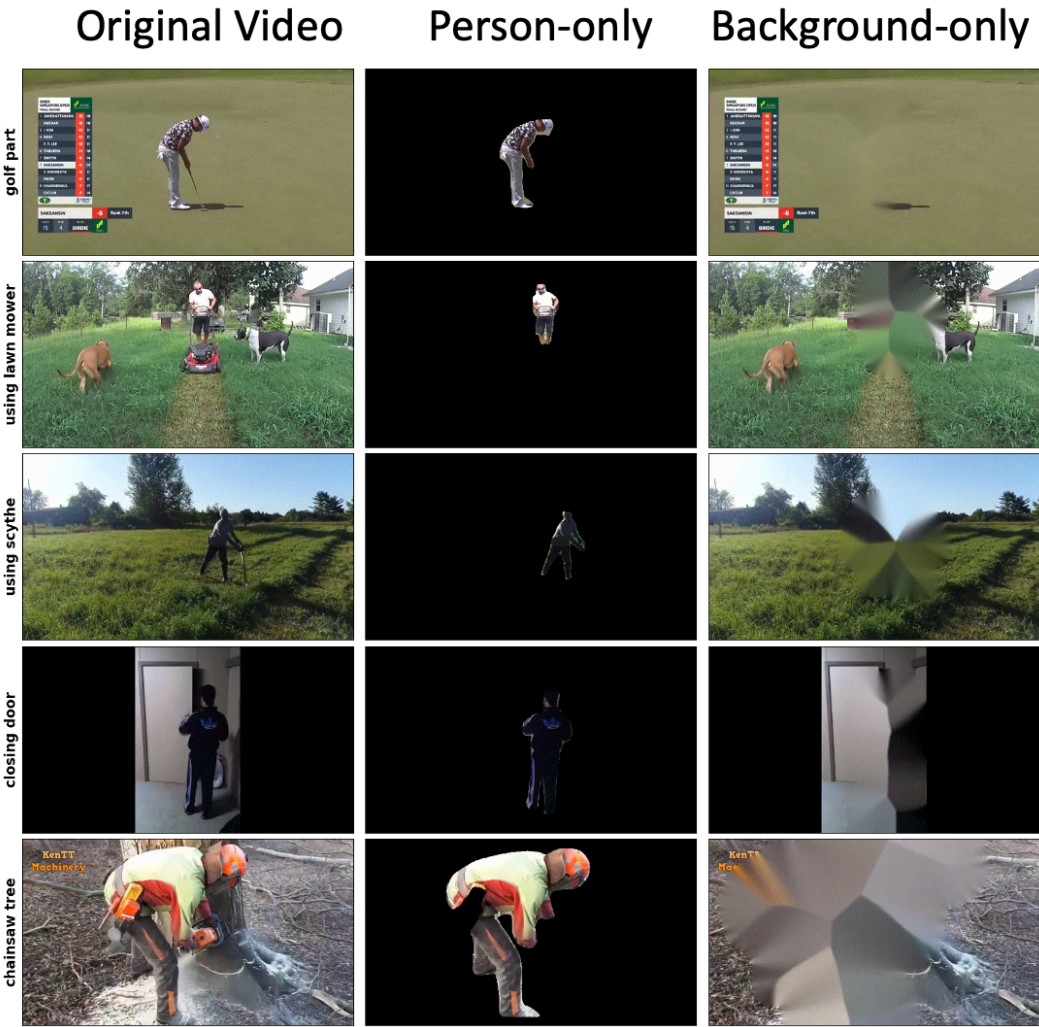

Figure S1: Example stimuli.

Table S1: Top 1 Accuracies (in percentage points) of networks compared.

| Network-name | ORIG | body-only | background-only |
|---|---|---|---|
| Baseline: frames: | 5.21 | 0.97 | 2.65 |
| Baseline: frames+flows: | 7.76 | 2.25 | 5.66 |
| DomainNet: frames: | 16.16 | 14.15 | 4.42 |
| DomainNet: frames+flows: | **24.9** | **25.85** | **5.83** |

arm_wave

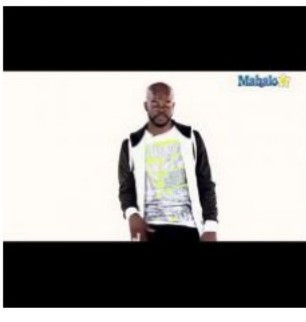

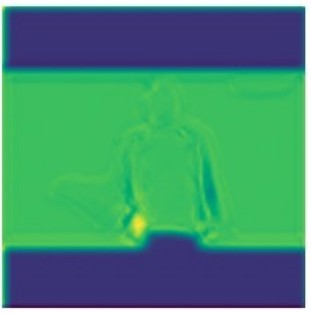

Figure S2: Example frame and corresponding flow used to train Baseline:frames+flows model

Table S2: Top 5 Accuracies (in percentage points) of networks compared.

| Network-name | ORIG | body-only | background-only |
|---|---|---|---|
| Baseline: frames: | 21.28 | 6.42 | 12.62 |
| Baseline: frames+flows: | 23.93 | 5.07 | 15.18 |
| DomainNet: frames: | 37.78 | 39.17 | **22.27** |
| DomainNet: frames+flows: | **52.89** | **55.39** | 19.08 |

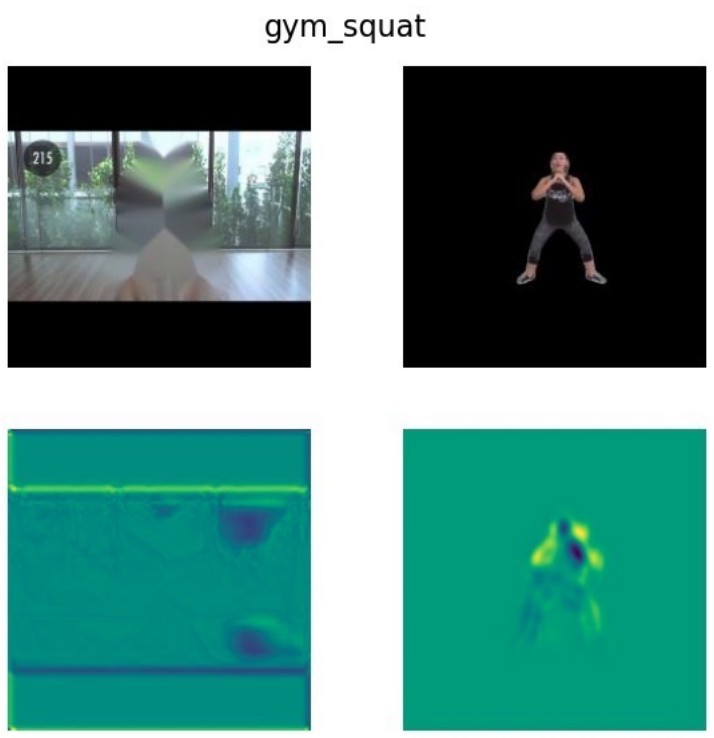

Figure S3: Example frames and corresponding flows used to train DomainNet:frames+flows model

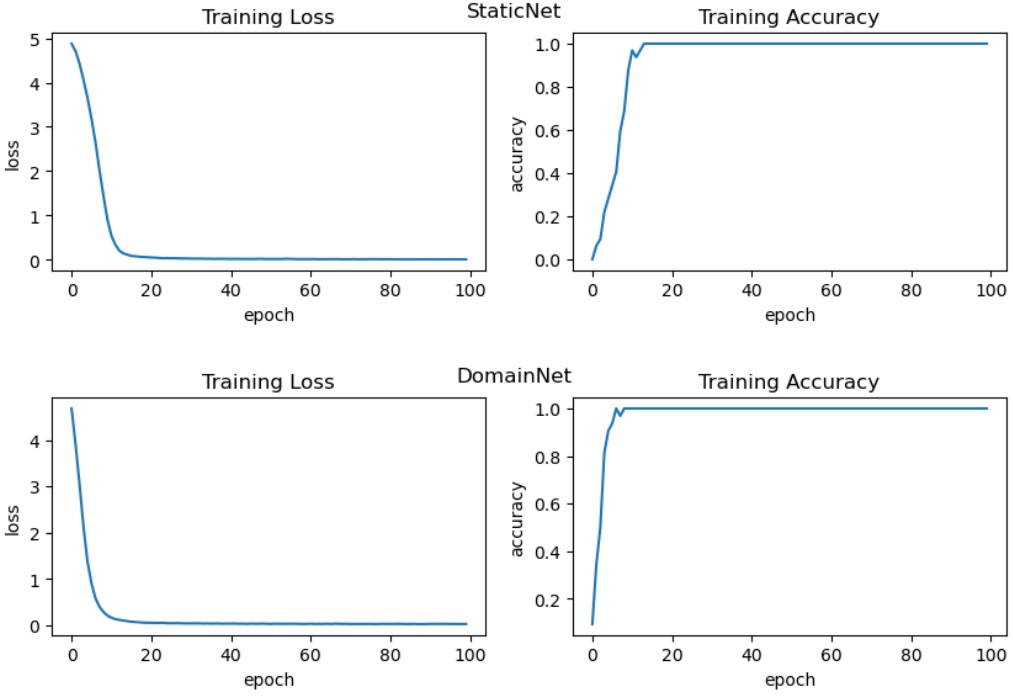

Figure S4: Training dynamics for networks tested. Training loss and training accuracy for both Baseline model and DomainNet converged in fewer than 20 epochs.

