# OpenReview forum: "Deep Models modelled after human brain boost performance in action classification"
_ICLR.cc/2024/Conference — Submitted to ICLR 2024_

### Official Review · Reviewer_W6Z2 · 2023-10-29

**Soundness:** 1 poor
**Presentation:** 1 poor
**Contribution:** 1 poor
**Rating:** 1
**Confidence:** 5

**Summary:**

The authors investigate how neural networks label action-recognition video frames and compare the neural network against human behavioral performance. They manipulate the stimuli to separate bodies and background and show that both neural networks and humans perform very well with full stimuli, body only or background only. Humans perform better with the body only compared to background only conditions. The authors propose an architecture that is loosely based on notions of modularity in the brain and this new architecture improves performance and matching to human behavioral data.

**Strengths:**

Building networks to recognize actions is of high importance to practical applications

Comparing how well networks perform to human performance is also of interest in terms of aligning machine and human visual capabilities.

**Weaknesses:**

The bottom line is that highly uncontrolled and bad datasets lead to spurious and uninterpretable results. This is the main challenge throughout.

In Fig. 1 bottom left, the authors claim that baseline models perform similarly well when tested with ORIG, body or BG. This is NOT what the results show. The results do not have error bars, let alone any minimally rigorous statistical analysis. From eyeballing the figure, it seems that ORIG>>BG>>Body.

The fact that humans can identify actions purely from the background frames with over 0.7 accuracy shows that
(1) The dataset is way too easy
(2) Background is a major confounding factor
(3) Time is not needed in such an easy task

As far as I understand, the proposed architecture is trained very differently from the baseline architectures. The proposed architecture is trained with Body-only stimuli and does better on Body-only stimuli, and it is trained on BG-only stimuli and does not perform better on BG-only stimuli (again, no error bars, no statistics, this is all from eyeballing). Training yields better performance in general.

It would be great to present actual results on how well Yolo v8 separates body and background.

The manuscript only has one main figure and minimal additional information that does not satisfy basic standards in the field. There are no error bars, there are no comparisons across multiple different models, no comparisons with different datasets, no ablations, no description of the effects of key variables like size, etc.

**Questions:**

A key aspect of action recognition is likely to be dynamics and time, which is not studied here.

Within the study of action recognition from frames, it would be useful to use rigorous datasets. If the authors are interested in the effect body and background, it would be important to rigorously control for basic variables like contrast, size, and multiple other confounds.

---

> ### Author Response · Authors · 2023-11-21
>
> Thank you for your review. Naturalistic datasets such as the HAA500 used in this manuscript are uncontrolled, however this does not invalidate our conclusions. The use of naturalistic datasets is very common in the field (see for example the Kinetics dataset). The use of such datasets makes it possible to train artificial neural networks to generalize across sources of variation that are encountered in the wild. Importantly, we used the same dataset to train both Baseline and DomainNet models. Therefore, differences between the two models cannot be due to differences in the datasets.The only difference between the training schemes was the architecture of the networks. As such, our results can only be explained by the difference between the architectures, with the two-stream architecture that separates body and background being advantageous over the single-stream architecture for the task of action classification.
>
> Dynamics and time do indeed contribute to action information and we did study the effects of both static and static plus dynamic information using optic flow. Those results are reported in Table 1 and are discussed in the Results section. Even when dynamic information is available, the model additionally benefits from separating background and body information (52.50% accuracy vs 75% accuracy), showing that dynamics alone does not account for our findings.
>
> In response to you other comments:
>
> "The fact that humans can identify actions purely from the background frames with over 0.7 accuracy shows that (1) The dataset is way too easy (2) Background is a major confounding factor (3) Time is not needed in such an easy task."
>
> 1) The task was easy for humans, but the baseline model nevertheless exhibited lower performance than both human participants and the proposed DomainNet model. There was no indication of a ceiling-effect that would suggest that the dataset was too easy.
>
> 2) That is correct, and was the motivation for the paper. This problem is not unique to HAA500, actions tend to be associated with particular backgrounds in naturalistic experience (e.g., “playing basketball” will most often happen in a basketball court). Therefore, it is important to build models that are robust to this. Results demonstrate that DomainNet was more robust to the confounding effect of background compared to the Baseline model.
>
> 3) “Time is not needed in such an easy task”. In contrast with this statement, our results show that including dynamic, time-dependent information improved the accuracy on our task compared to using only static information (see Table 1). Specifically, when processing the original videos (ORIG), DomainNet(frames+flows) – that uses both static frames and optic flow (which are time-dependent) – obtained an accuracy of 75%, while DomainNet(frames) – that does not use time-dependent information – only obtained an accuracy of 66.25%. This was also the case for processing videos showing only the body: DomainNet(frames+flows) obtained an accuracy of 73.75%, while DomainNet(frames) obtained an accuracy of only 62.5%. These results show that time-dependent information (optic flow) was indeed helpful to perform the task, and that it led to improvements in action recognition accuracy.
>
> “It would be great to present actual results on how well Yolo v8 separates body and background.”
>
> These are available in figure S1.
>
> “As far as I understand, the proposed architecture is trained very differently from the baseline architectures. The proposed architecture is trained with Body-only stimuli and does better on Body-only stimuli, and it is trained on BG-only stimuli and does not perform better on BG-only stimuli (again, no error bars, no statistics, this is all from eyeballing). Training yields better performance in general.”
>
> The only difference between the training of the Baseline network and the DomainNet was that baseline models were trained with body and background information mixed within the same stream”, while DomainNet models were trained with body and background separated into different streams, and separate loss functions were used for each stream to drive each of the streams to perform as accurately as possible. The loss function used for both the DomainNet and Baseline models was the cross-entropy loss. Both the DomainNet and the Baseline models were trained for 100 epochs, with 500 batches per epoch and 32 stimuli per batch. Both the DomainNet and the Baseline models were trained using stochastic gradient descent (SGD), with learning rate of 0.001 and momentum of 0.9. Therefore, all aspects of training were identical for the DomainNet and the Baseline models, except from the use of separate losses for the two streams which was the key manipulation of interest.

---

> > ### Comment · Reviewer_W6Z2 · 2023-11-23
> > **I remain highly skeptic**
> >
> > The authors failed to address the questions raised in the review, insisting on statements without error bars, statistical analyses or any minimal scientific rigor.
> >
> > I won't reiterate all the questions. Just to highlight a few examples:
> >
> > As noted in the review, time is not needed in such an easy task. From eyeballing the results (the authors still have not provided any rigorous way of evaluating the results), adding time hurts performance for DomainNet for the background only, and helps with the body-only and ORIG conditions. Focusing on body-only, which is the one that most helps for the time argument, chance is 20%, just using frames, performance goes from 20% to 62.50% (42.5% improvement with respect to chance) showing that time is indeed not needed for the main boost in performance. Adding temporal information in the form of optic flow brings a boost of 11% (about 1/4 of the effect due to the frame itself). Again, the gain in performance with respect to chance is mostly driven by each frame. To study the role of time, one would like to design a dataset where frame information is truly insufficient for action recognition. One example of this is the case of recognizing action from simple light sources from actors in a dark background. Using such a controlled dataset could bring insights into how to incorporate temporal information for action recognition. To be clear, I think that temporal information is likely to be extremely interesting to rigorously study for action recognition and I would love to see rigorous work in this direction.
> >
> > Indeed, the baseline models were trained differently from the proposed architecture. Why is it surprising that if you train with body only, the model will get better performance in the body-only condition? Presumably, the authors could divide the image into 4 quarters, train with the upper left quarter, and show improved performance in the upper left corner. None of this has anything to do with action recognition.

---

### Official Review · Reviewer_JzJq · 2023-10-31

**Soundness:** 2 fair
**Presentation:** 1 poor
**Contribution:** 1 poor
**Rating:** 1
**Confidence:** 4

**Summary:**

This work begins by examining the similarities and differences between humans and deep neural networks in terms of action recognition. It demonstrates that a deep neural network trained with cross entropy on the entire video cannot perform action recognition when background information is omitted from the training data. In contrast to this, human subjects are capable of identifying activities solely from the body information. This highlights that DNN trained for action recognition incorrectly balances the body and background information present in the video data. In order for the deep neural network to exclusively distinguish actions coming from the body, the authors suggested using two different backbones, one for the body and one for the background. In addition to this, they implemented a loss function that was more complex and yet nevertheless compatible with their category-selective design. As a consequence of this, they demonstrated that a body-background separated backbone may produce an action recognition pattern that is comparable to the pattern seen in human participants, albeit with a significantly lower level of accuracy.

**Strengths:**

The authors try to optimize deep neural works towards reproducing action recognition patterns observed in human subjects.

**Weaknesses:**

As the authors already included in the related works, having separate streams for different information in video is not new. For example, an early work on dynamic texture processing used two separate backbones for “appearance” (the scene) and “dynamic” (the optic flow). Their loss function also combines the matching of both appearance and dynamic features. It is possible that the L_{combined} here is new. However, the authors do not include any details on how they define L_{body}, L_{background} or L_{combined}. If the only difference this work has with other work is its usage of L_{combined} (I guess this is the cross entropy between predicted frames vs. the true frames), the novelty is very limited.



Tesfaldet et al 2017 Two-Stream Convolutional Networks for Dynamic Texture Synthesis



This work needs a much more developed result section to fit as an ICLR manuscript. Its current format has one result. This result is not surprising given previous literature. I would encourage the authors to include a more detailed investigation of the proper loss functions, what predictive features are being used in humans to perform action recognition, etc. These extensions may strengthen this paper.

**Questions:**

Does the background contain any information about the actions in the video? If not, I hope the authors illustrate better why the background information should be used at all for action recognition. Would it be desirable that a neural architecture should focus on the “action” component of the video to perform action recognition?

Which component of the loss function contributes the most when body-only information is being used? Or when background-only information is being used?

---

> ### Author Response · Authors · 2023-11-21
>
> Thank you for your feedback. Crucially, while previous work used separate streams for static and dynamic information (e.g. Tesfaldet et al 2017) we know of no previous work that segmented a single information modality (e.g. static information) to be processed by separate streams. This idea takes root in domain-selectivity literature (Caramazza & Shelton, 1984) which we demonstrate to be beneficial for the field of machine learning.
>
> The loss function used to train the DomainNet model was kept purposefully simple and was a combination of three losses: background stream loss, body stream loss and a combined loss. Body stream loss is a cross-entropy loss (prediction of action category from input frame) using body-segmented frames, while background loss is a cross-entropy loss using context only (body in-painted) frames. The combined loss is a sum of the previous two losses. While we agree that additional research on better loss functions could be beneficial, we used this relatively straightforward arrangement of losses in order to clearly demonstrate that performance gains are due to having separate information streams for body and background.
>
> Background information does indeed contain information about the action class (e.g. playing baseball tends to happen on a baseball field etc.). This is further supported by our findings that both neural networks and human observers can accurately predict action category when body information is removed via inpainting (Figure 1). The challenge is that models can learn to overly rely on this contextual information and ignore body-pose related information during training leading to lower model performance overall (a conclusion supported by our results). Only using body-pose information would result in higher performance than only using background information, however an even better option is to combine both (as we did in DomainNet). As you can see in figure 1 and table 1, combining both body and background information results in higher accuracies than using either source alone. Additionally, this mirrors human behavioral results, suggesting that humans too combine both background and body information when understanding actions.

---

### Official Review · Reviewer_df1j · 2023-11-01

**Soundness:** 3 good
**Presentation:** 2 fair
**Contribution:** 2 fair
**Rating:** 3
**Confidence:** 4

**Summary:**

This paper takes insights from neuroscience and builds a model that processes body and background in images separately, aiming to improve the performance of action recognition.

**Strengths:**

This paper propose novel ideas of incorporating inductive bias from neuroscience in building artificial neural networks and shows that it does improve the performance of action recognition when compared with a baseline network. The paper is well-written and very clear. The human dataset collected in this paper is also valuable and should be perhaps incorporated into action recognition benchmarks.

**Weaknesses:**

This paper is a rudimentary effort in showing incorporating certain inductive bias from neuroscience could potentially help with artificial networks in certain tasks. However for the scope of the conference, I think the lack of comparison to state-of-art models as well as insights on how to even combine this inductive bias with state-of-art models makes this paper not suitable for application and making real impact on the task of action recognition. It is also not entirely true to assume that state-of-art model, which is much more complicated than a ResNet50 network does not implicitly extract information from the background and body when recognizing action.

**Questions:**

Discussion of how to incorporate this into state-of-art models is recommended.

---

> ### Author Response · Authors · 2023-11-21
>
> Thank you for your comments. The goal of our work is primarily to advance models of action representations in the human brain, therefore we were interested in testing whether incorporating inductive biases from neuroscience can lead to models with more human-like behavior. It is possible that our goals do not align sufficiently with the typical content of ICLR. We agree with your assessment that our work does not test whether other state-of-the-art action recognition models can achieve similar performance: building the most accurate model was not  our aim. We nonetheless believe that this work can offer insights for artificial networks, in at least two ways. First, this work motivates a more systematic study of the contributions of body and background information in state-of-the-art models. More generally, it motivates the study of models that process different object categories in separate streams, not only in the context of action recognition, but also in the context of other applications such as video prediction. Second, this work could benefit applications for which large models are not viable due to computational constraints (e.g. Internet of Things devices).

---

### Meta-Review · Area_Chair_hH2z · 2023-12-05

**Metareview:**

The reviewers unanimously rejected the paper because of a lack of experimental rigor. The AC agrees and recommends the paper to be rejected.

**Justification For Why Not Higher Score:**

The reviewers unanimously rejected the paper because of a lack of experimental rigor.

**Justification For Why Not Lower Score:**

NA

---

### Decision · Program_Chairs · 2024-01-16

Reject